# A High-Proton Conductivity All-Biomass Proton Exchange Membrane Enabled by Adenine and Thymine Modified Cellulose Nanofibers

**DOI:** 10.3390/polym16081060

**Published:** 2024-04-11

**Authors:** Chong Xie, Runde Yang, Xing Wan, Haorong Li, Liangyao Ge, Xiaofeng Li, Guanglei Zhao

**Affiliations:** 1State Key Laboratory of Pulp and Paper Engineering, School of Light Industry and Engineering, South China University of Technology, Wushan Road, Guangzhou 510641, China; 201810107072@mail.scut.edu.cn (C.X.); 202021029251@mail.scut.edu.cn (R.Y.); 202121030223@mail.scut.edu.cn (X.W.); 202220128588@mail.scut.edu.cn (H.L.); ligeliangyao@mail.scut.edu.cn (L.G.); 2School of Food Science and Engineering, South China University of Technology, Wushan Road, Guangzhou 510641, China

**Keywords:** tempo-oxidized cellulose nanofibers, proton exchange membrane, adenine, thymine

## Abstract

Nanocellulose fiber materials were considered promising biomaterials due to their excellent biodegradability, biocompatibility, high hydrophilicity, and cost-effectiveness. However, their low proton conductivity significantly limited their application as proton exchange membranes. The methods previously reported to increase their proton conductivity often introduced non-biodegradable groups and compounds, which resulted in the loss of the basic advantages of this natural polymer in terms of biodegradability. In this work, a green and sustainable strategy was developed to prepare cellulose-based proton exchange membranes that could simultaneously meet sustainability and high-performance criteria. Adenine and thymine were introduced onto the surface of tempo-oxidized nanocellulose fibers (TOCNF) to provide many transition sites for proton conduction. Once modified, the proton conductivity of the TOCNF membrane increased by 31.2 times compared to the original membrane, with a specific surface area that had risen from 6.1 m²/g to 86.5 m²/g. The wet strength also increased. This study paved a new path for the preparation of environmentally friendly membrane materials that could replace the commonly used non-degradable ones, highlighting the potential of nanocellulose fiber membrane materials in sustainable applications such as fuel cells, supercapacitors, and solid-state batteries.

## 1. Introduction

Proton exchange membrane fuel cells (PEMFCs) have been widely acknowledged for their environmental friendliness and high power density, making them suitable for various energy applications, including mobile, fixed, and portable uses [1,2,3,4,5,6]. As a crucial component, the proton exchange membrane (PEM) plays a decisive role in the performance and service life of fuel cells. Nafion membrane, being the most used proton exchange membrane, exhibited good stability under moderate temperature conditions, high proton conductivity, and favorable processability [7,8,9,10]. It possessed approximately 4 nm micro-water environment channels and abundant negatively charged sulfonic acid groups, ensuring very high ion exchange capacity and proton conductivity [11,12,13]. However, its larger pore structure led to slightly higher methanol permeability, potentially decreasing the coulombic efficiency of the battery [14,15]. Moreover, the sensitivity of sulfonic acid groups to pH necessitates an acidic operating environment [16]. To address these drawbacks, the development of proton exchange membranes based on cationic functional group polymers like amines, purines, imidazoles, and pyridines drew considerable research interest [17,18,19,20].

With growing concerns over environmental and human health risks, alongside the drive toward carbon neutrality, the design of proton exchange membranes from natural polymers like cellulose and chitosan gained widespread attention [21,22,23,24]. Compared to synthetic polymers, natural polymers offered advantages like abundant availability, biodegradability, and biological compatibility. Our team had previously created proton exchange membranes with low initiation temperatures, broad temperature ranges, low methanol permeability, and high proton conductivity by modifying cellulose nanocrystals (CNC) and chitosan with adenine, phosphate, and sulfonic acid groups [25,26,27]. This demonstrated the viability of CNC and chitosan as substrates for proton exchange membranes and elucidated the transfer and transition mechanisms of adenine groups in proton conduction. However, prior studies had not explored the impact of the nanoporous structure of proton exchange membranes on proton conductivity. In fact, many researchers have reported that by grafting DNA segments or base groups on the surface of nanoparticles or nanofibers, they can be induced to aggregate according to certain rules [28,29,30,31,32,33,34]. Therefore, we integrated adenine (A) and thymine (T) into nanocellulose fibers, manipulating pH to induce aggregation, enhance membrane porosity, and enlarge pore sizes. This enhancement of micro-water channels facilitated proton conduction and transition, thereby improving the proton conductivity of the membranes. Adenine and thymine are easily solvated under hot, acidic, or alkaline conditions and dispersed uniformly in water. However, at neutral room temperature, they tended to aggregate. Under acidic conditions, these bases were grafted onto the surface of tempo-oxidized nanocellulose fibers (TOCNF). Following dialysis to neutrality and concentration by evaporation, the fiber surfaces modified with adenine and thymine were preferentially aggregated compared to unmodified areas. This aggregation effectively introduces paired bases into orderly pores, regulating pore size and increasing pore volume.

In this work, we developed a simple and environmentally friendly method to prepare TOCNF-based proton exchange membranes. The process involves two steps: first, adenine and thymine are grafted onto the carboxyl groups on the surface of TOCNF through 1-(3-dimethylaminopropyl)-3-ethylcarbodiimide hydrochloride (EDCI)-N-hydroxy succinimide (NHS) catalysis at room temperature. Subsequently, the adenine-modified TOCNF and thymine-modified TOCNF are mixed evenly, dialyzed to neutrality, and slowly dried in a constant temperature and humidity chamber at 60 °C to form a membrane, named TOCNF-AT (Figure 1). The atomic force microscope (AFM) and scanning electron microscope (SEM) characterize the surface morphology of the proton exchange membrane, indicating that the nanofibers in TOCNF-AT pre-aggregate before film formation. Compared with the unmodified TOCNF membrane, the specific surface area of the modified membrane significantly increases, with the main pore size expanding from 1.49 nm to 2.52 nm. The room-temperature proton conductivity increases from 0.0017 S cm^−1^ to 0.053 S cm^−1^. Although the dry strength decreases after modification, the wet strength increases, which is beneficial for application in the humid environment of fuel cells. After soaking for one week, the proton conductivity of the TOCNF membrane remains essentially unchanged. Importantly, we introduced a new, green, simple technique to control the aggregation behavior of nanofibers by adjusting hydrophilicity and hydrophobicity, thereby preparing a high-conductivity proton exchange membrane, which reduces costs and promotes the wide application of fuel cell technology.

## 2. Materials and Methods

### 2.1. Experimental Materials

The 2,2,6,6-tetramethylpiperidine-1-oxyl radical (TEMPO), cotton cellulose fiber, 1-(3-Dimethylaminopropyl)-3-ethylcarbodiimide hydrochloride (EDCI), N-hydroxy succinimide (NHS), adenine (A), and thymine (T) were procured from Sigma-Aldrich (St. Louis, MO, USA). 

### 2.2. Preparation of TOCNF and TOCNF-AT Proton Exchange Membranes

Catalytic amounts of TEMPO and NaBr were dissolved in an aqueous solution of sodium hydroxide with a pH value of 10–11. The degreasing cotton fiber with a cellulose mass concentration of 1 wt.% was added, followed by the addition of NaClO as the main oxidant to start oxidation. After 6 h of reaction at room temperature, the TEMPO-oxidized cellulose nanofiber TOCNF was obtained by repeated dialysis with deionized water. The carboxyl content of TOCNF was determined to be 1.3 mmol/g by titration.

In PBS buffer (pH = 5), TOCNF was added and stirred overnight. Then, a predetermined amount of adenine and NHS was added to the TOCNF dispersion and stirred for 30 min. An equal molar amount of EDCI was then added to the reaction solution and allowed to react for 24 h at room temperature. Due to the low water solubility of adenine at room temperature, a small amount of it needed to be added repeatedly. The same applied to the thymine group. The reaction products of the adenine group and the thymine group were then mixed and stirred at room temperature for 24 h. Finally, the product was purified using PBS buffer (pH = 5) and deionized water by dialysis (MWCO: 8000–14,000 Da). The water solubility of adenine and thymine is sensitive to pH, and direct dialysis with deionized water could lead to the precipitation of unreacted adenine and thymine in the dialysis bag due to an increase in pH. Therefore, PBS buffer was needed to remove unreacted adenine and thymine, as well as EDCI and NHS, followed by dialysis with deionized water to remove the PBS. During dialysis, the dialysis solution was measured by ultraviolet-visible light to ensure that free bases and other catalysts were completely removed, resulting in a dispersion of TOCNF-AT. The TOCNF and TOCNF-AT dispersions were placed in polytetrafluoroethylene molds, respectively, and dried in a constant temperature and humidity drying oven at 60 °C for 24 h. Then, they were dried in a vacuum drying oven at 50 °C for 24 h, resulting in two proton exchange membranes.

### 2.3. Characterizations

To investigate the chemical reaction between TOCNF and adenine and thymine, the samples were analyzed using Fourier transform infrared spectroscopy (FTIR) and attenuated total reflection (ATR) prestige-21 (Shimadzu, Kyoto, Japan). The FTIR-ATR spectrum was taken from 600 to 4000 cm^−1^. The morphology of the samples was examined using a scanning electron microscope (SEM) (SU8000, HITACHI, Ichihara, Japan) and an atomic force microscope (AFM) (Dimension Icon, Bruker, Billerica, MA, USA). The acceleration energy of all samples was maintained at 5 kV. The crystal structures of cotton cellulose fibers, TOCNF, and TOCNF-AT were observed by X-ray diffraction (XRD) spectroscopy. The XRD pattern was taken at room temperature (25 ± 3 °C) using a Rotaflex RT300 mA, Shimadzu, Osaka, Japan, with an angle (2θ) range of 5 ≤ 2θ ≤ 60°. The mechanical properties of the prepared TOCNF and TOCNF-AT films were evaluated using a universal testing machine (UTM), Tensilon RTC 250A, A&D Company Ltd., Osaka, Japan. Surface area and pore size distribution are tested by BET (ASAP, Micromeritics, Norcross, GA, USA).

### 2.4. Proton Conductivity (σ)

Electrochemical impedance was measured using a CHI 760D electrochemical workstation (SUIOU, Shanghai, China) at a frequency range of 0.001 Hz–105 Hz at an alternating current amplitude of 0.5 V. The three electrodes used were the working electrode, the reference electrode, and the counter electrode. The working electrodes were a customized TOCNF membrane and a TOCNF-AT membrane held by a dedicated polytetrafluoroethylene fixture. The reference electrode was a silver chloride reference electrode filled with potassium chloride, and the platinum piece served as the counter electrode. All electrodes were fixed with an orifice plate to ensure parallelism with the membrane. The proton conductivity of the samples was calculated using the following equation:$$\sigma =\frac{L}{RS}$$
where *σ* is the proton conductivity (S cm^−1^), *L* is the length (cm) of the membrane, *S* is the cross-sectional area of the membrane (cm^2^), and *R* is the resistance calculated from the electrochemical impedance spectra (Ω).

## 3. Results and Discussion

### 3.1. Fourier Transform Infrared Absorption Spectroscopy Analysis

Cotton fibers underwent TEMPO oxidation to obtain TOCNF, wherein the hydroxyl group at the C6 position was oxidized to a carboxyl group. The FTIR spectrum (Figure 2) of the resultant TOCNF showcased a stretching vibration peak of the C=O bond at 1720 cm^−1^, a peak absent in the spectrum of the original cotton fibers. Upon undergoing a reaction catalyzed by EDCI-NHS, adenine and thymine reacted with TOCNF to form a tertiary amide structure, with the peak at 1641 cm^−1^ corresponding to the stretching vibration peak of the C=O bond tertiary amide. Owing to the attachment of the tertiary amide structure to electron donor groups, a slight blue shift was observed when compared to the primary amide structure. In terms of base complementarity, the following two typical forms exist: Watson–Crick pairing and Hoogsteen pairing [35,36,37]. Studies from the past have demonstrated that in the liquid phase, Hoogsteen pairing tends to be predominant, exhibiting a distinctive infrared absorption peak. In the instance of adenine and thymine forming a hydrogen bond structure via Hoogsteen pairing, the corresponding stretching vibration peaks were at 3197 cm^−1^ and 3395 cm^−1^ [38]. However, in an unfortunate turn of events, these peaks were not observed in the FTIR spectrum of the TOCNF-AT. This absence of peaks suggested that the adenine and thymine groups grafted onto the surface of TOCNF-AT did not aggregate significantly in alignment with Hoogsteen pairing. The lack of aggregation could be attributed to the rigidity inherent to the TOCNF fibers, coupled with the reality that the modified base groups comprised only a minor portion of the fiber surface. Consequently, it became challenging to ensure adequate free collisions between adenine and thymine when the nanofibers were concentrated and aggregated, resulting in a more random nature of the aggregation. 

### 3.2. X-ray Diffraction Analysis

Figure 3 displays the typical XRD pattern of type I cellulose, featuring diffraction peaks at 16.68° and 22.4° in 2θ. The crystallinity of cotton cellulose fiber was 66.8%, the crystallinity of TOCNF was 55.3%, and the crystallinity of TOCNF-AT was 51.4%. As cotton fibers underwent nanofiberization, the crystallinity of cellulose was reduced. This reduction was attributable to the fact that during the TEMPO oxidation process, the cellulose fibers were initially immersed in a sodium hydroxide solution, leading to the disruption of some of the hydrogen bonds within the cellulose. Subsequently, carboxylation of the hydroxyl group at C6 ensued, causing a degree of disruption to the original regular hydrogen bond network. Nonetheless, the crystallinity continued to exhibit a decrease even during the TOCNF modification with adenine and thymine. This phenomenon could likely be due to the esterification of the carboxyl groups by EDCI, modifying adenine and thymine, which possess a comparatively low surface affinity for cellulose. These components, acting as “impurities” on the surface of cellulose, interfered with the establishment of regular hydrogen bond connections between fiber surfaces as the fibers aggregated. However, from the standpoint of proton exchange membrane functionality, lower crystallinity signifies an abundance of proton migration pathways. This feature was conducive to the enhancement of proton conductivity.

### 3.3. Morphological Characterization

High-resolution scanning electron microscopy (SEM) and atomic force microscopy (AFM) were utilized to examine the surface structure of TOCNF prior to and following modification. By analyzing the AFM images with ImageJ software (1.8.0), the fiber diameter distribution of TOCNF was determined to range between 20 and 100 nm (Figure 4a), whereas the fiber diameter distribution of TOCNF-AT was found to range between 50 and 300 nm (Figure 4b). This phenomenon indicated that TOCNF-AT exhibited non-uniform random aggregation. Based on the aggregation behavior of adenine and thymine in neutral water at room temperature, this phenomenon could be attributed to the preferential aggregation of parts of the fiber surface modified with adenine and thymine during the neutralization and evaporation concentration processes, which induces the ordered parallel aggregation of nanofibers. From the SEM images, it was observed that TOCNF (Figure 4c) aggregated more densely compared to TOCNF-AT (Figure 4d). This could be due to the higher rigidity of TOCNF-AT, which results from the parallel aggregation of several TOCNF in contrast to the thinner single cellulose nanofiber, as well as non-uniform aggregation factors. In Figure 4d, it was also noted that thicker fibers consisted of thinner fiber branches.

Whether it is the surface-loading of adenine or thymine, TOCNF will aggregate during evaporation and concentration. If no base group is modified, during the evaporation and aggregation processes, TOCNF will first be connected by water bridges. As the free water evaporates, the distance between nanofibers is further reduced due to surface tension, forming hydrogen bonds between hydroxyl groups. However, after introducing adenine and thymine onto the surface of cellulose, they play a role in inducing the aggregation of TOCNF. Compared with unmodified TOCNF, although both have aggregated, the former occurs evenly while the latter is affected by base groups and occurs unevenly. During the evaporation and concentration process of TOCNF-AT, adenine and thymine groups may preferentially aggregate, and the aggregated parts hinder the formation of hydrogen bonds between nearby cellulose hydroxyl groups, increasing the porosity between celluloses. At the same time, the occurrence of uneven aggregation will further lead to an increase in porosity and specific surface area. However, it is indeed difficult to observe the changes in the pore size and specific surface area of TOCNF membranes and TOCNF-AT membranes from AFM and SEM images. These changes require BET data as direct evidence. Since the AFM and SEM equipment used in this work do not have sufficient lateral and vertical resolution to support the observation of small changes below 5 nm, it is not practical to directly observe the changes in specific surface areas and micropores from morphological characteristics.

### 3.4. BET Analysis

The BET-specific surface area of the TOCNF membrane registered a mere 6.1 m²/g, as characterized by N2 adsorption. However, following the modification with adenine and thymine and the subsequent induced aggregation, the BET-specific surface area of the TOCNF-AT membrane escalated significantly to 86.5 m²/g (Figure 5a). The maximum pore size of the TOCNF membrane stood at 1.5 nm (Figure 5b), whereas the maximum pore size of the TOCNF-AT membrane widened to a range of 2.6–3.1 nm (Figure 5c). The augmentation in specific surface area was ascribed to the heterogeneous aggregation of the modified cellulose nanofibers and to the dilation of aggregate spaces between cellulose molecules by the adenine and thymine groups on the cellulose surface. In fact, during the drying and concentration process, both TOCNF and TOCNF-AT underwent aggregation. However, the difference lies in the fact that the aggregation of TOCNF was uniform while that of TOCNF-AT was not. During the drying and concentration process of TOCNF, water bridges were formed between TOCNF molecules, and as the free water evaporated, the distance between TOCNF molecules was further reduced due to surface tension, forming hydrogen bonds between hydroxyl groups. This process occurred uniformly. In the concentration process of TONCF-AT, due to the induction effect of adenine and thymine, the surface part of the cellulose modified by base groups expelled water molecules upon reaching a certain concentration, resulting in aggregation. However, only a carboxyl content of 1.3 mmol/g made the distribution of adenine and thymine on the surface of the nanofibers very uneven, leading to an uneven aggregation as well, thereby increasing the specific surface area. The amplified specific surface area implies the presence of an increased number of microchannels available for proton conduction and transition, which is advantageous for proton conductivity. The enlarged pores could mitigate the energy required for transportation, albeit still falling short of the approximate 4 nm pore size delineated in the literature for Nafion membranes.

### 3.5. Dry and Wet Strength Tests of TOCNF and TOCNF-AT Membranes

The dry strength of TOCNF and TOCNF-AT was assessed using samples that had been completely dried to eliminate free water. Conversely, the wet strength was evaluated using samples derived by immersing the dry films of both materials in deionized water at room temperature for one hour, followed by the removal of surface moisture with filter paper. The findings of these tests are illustrated in Figure 6. The tensile strength of TOCNF’s dry film registered at 77.8 MPa, and its elastic modulus stood at 3.4 GPa. In contrast, the dry film of TOCNF-AT displayed a tensile strength of 47.1 MPa and an elastic modulus of 3.8 GPa. The decrease in tensile strength for the modified nanofiber, TOCNF-AT, was significant due to its comparatively looser structure and larger pores. However, the nearly unchanged elastic modulus was attributed to the rigid bonding between adenine and thymine. Upon wetting, the tensile strength of TOCNF diminished to 2.1 MPa, and its modulus was reduced to 0.27 GPa. Conversely, the tensile strength and elastic modulus of the modified TOCNF-AT were higher than those of the unmodified version, recorded at 2.8 MPa and 0.48 GPa, respectively. The enhanced wet strength of the modified cellulose nanofibers can be attributed to the lower hydrophilicity of adenine and thymine compared to cellulose under neutral conditions at room temperature, resulting in fewer structures prone to swelling. This relatively higher wet strength of the TOCNF-AT proton exchange membrane, along with its capability to resist deformation under high humidity conditions, proves advantageous for applications in fuel cells.

### 3.6. Proton Conductivity Test

The assessment of proton conductivity emerged as the paramount characteristic for gauging the applicability of proton exchange membranes in devices like fuel cells. Owing to the presence of carboxyl groups on the surface of TOCNF, its proton conductivity generally far surpasses that of other cellulose materials [39,40,41]. Nonetheless, it lagged commercial Nafion membranes by more than an order of magnitude. Employing the alternating current impedance method (ESI), we determined the proton conductivity of the TOCNF-AT membrane at room temperature to be 0.053 S cm^−1^, marking an approximate increase of 31.2 times over that of TOCNF. Following a seven-day immersion in deionized water, the proton conductivity of TOCNF-AT showed negligible degradation (Figure 7). At lower temperatures, the proton conductivity of the TOCNF-AT proton exchange membrane approached that of the commercial Nafion117 membrane [11]. This phenomenon was credited to the abundant proton donors and acceptors found on adenine and thymine, coupled with the effective nano-water channels that resulted from non-uniform aggregation. The stability of the proton conductivity of TOCNF-AT is attributed to the non-hydrophilicity of adenine and thymine under neutral conditions at room temperature. This property ensures the stability of the structure and pores of the proton-conducting membrane in water. Compared to the previous research results of the team, the proton conductivity of adenine modified on the surface of chitosan under the same conditions was only 0.0309 S cm^−1^, which was nearly 40% lower than the research in this work [27]. The site of adenine modification on the surface of chitosan was the amino group, while the site of modification on the surface of TOCNF was the carboxyl group. Although imine structures and amide structures were formed, respectively, the chemical structures of other parts were similar. Even with the higher amino acid coverage of chitosan, its achievable grafting rate of adenine was higher. With the advantage of chemical structure, the proton conductivity rate was still lower than that in this work. Therefore, the significant difference in proton conductivity values might come from the contribution of nanochannels in TOCNF-AT proton exchange membranes. To further improve the proton conductivity of TOCNF-AT, increasing the carboxyl content on the surface of TOCNF and increasing the loading rate of base groups by increasing carboxyl groups could be achieved.

## 4. Conclusions

This study employed a straightforward and environmentally friendly methodology to fabricate a proton exchange membrane characterized by high proton conductivity, biodegradability, and eco-friendliness, utilizing the heterogeneous aggregation of TOCNF surface-modified by base groups. Through this innovative approach, the membrane not only acquired a high specific surface area and larger pore size but also manifested an enhanced wet strength and modulus. By incorporating base groups onto the surface of TOCNF, the preparations provided a multitude of proton jump sites and donor receptors, enhancing proton conduction and significantly sculpting rich micro-water channels in the membrane. These advancements facilitated the inventive integration of two distinct mechanisms. With an eye toward a sustainable future for proton exchange membrane fuel cells (PEMFCs), the newly developed membrane may emerge as an economically viable rival to current cost-intensive PEM technologies that rely on metals or non-degradable materials. This research paved a new path for the preparation of environmentally friendly proton exchange membrane materials and provided a new design concept for improving proton conduction efficiency. From the perspectives of process technology and material surface groups, the coordinated design of the structure of the proton exchange membrane meets the functional requirements of fuel cells. This further enhances the potential of natural materials in energy storage applications.

## Figures and Tables

**Figure 1 polymers-16-01060-f001:**
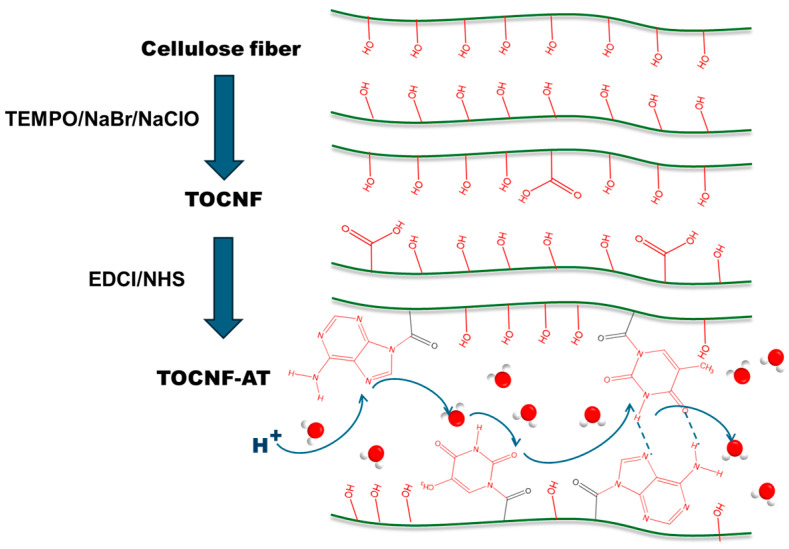
Schematic diagram of cellulose nanofiber modification and proton conduction.

**Figure 2 polymers-16-01060-f002:**
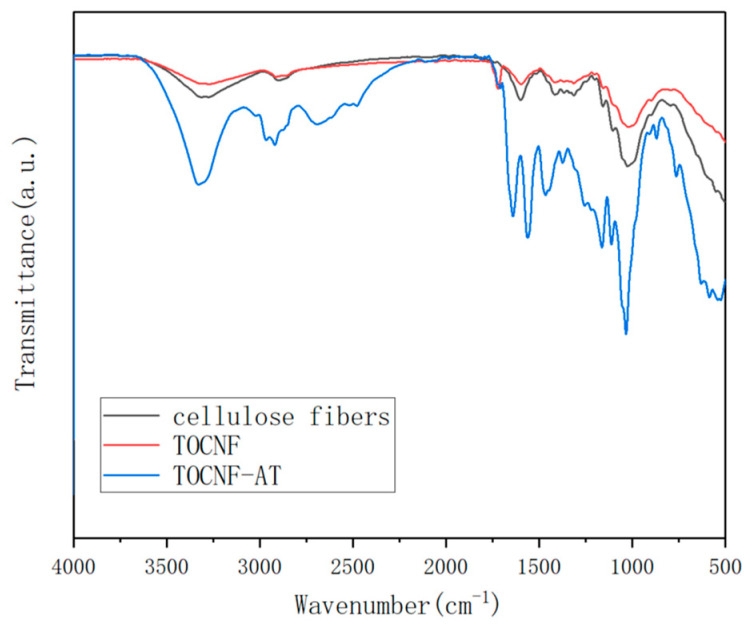
FTIR spectra of cellulose fibers, TOCNF and TOCNF-AT.

**Figure 3 polymers-16-01060-f003:**
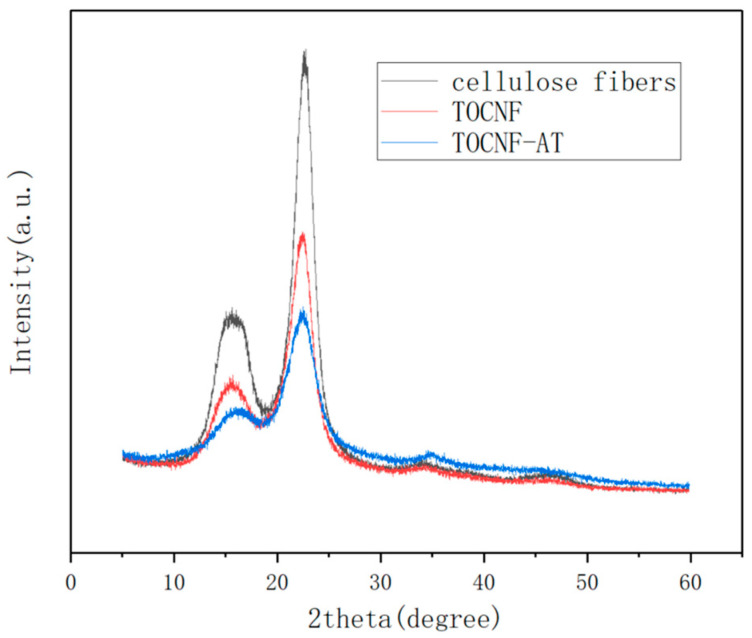
X-ray diffraction (XRD) spectra of cellulose fibers, TOCNF and TOCNF-AT.

**Figure 4 polymers-16-01060-f004:**
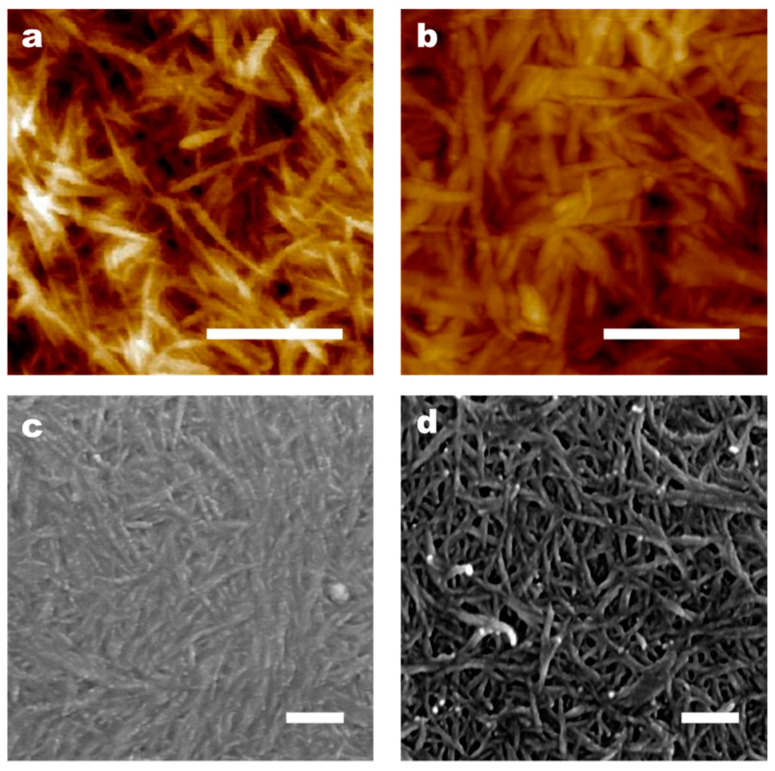
AFM image of the membrane: (**a**) TOCNF and (**b**) TOCNF-AT. SEM image of the membrane: (**c**) TOCNF and (**d**) TOCNF-AT. The white scale bar in the figure is 200 nm.

**Figure 5 polymers-16-01060-f005:**
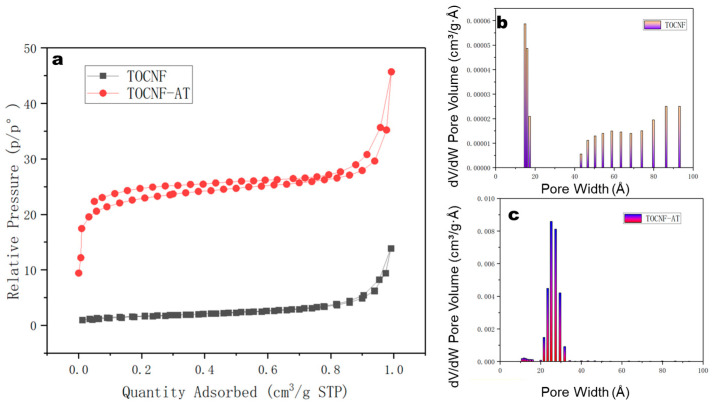
(**a**) Adsorption and desorption curves of nitrogen on TOCNF and TOCNF-AT; pore size distribution of (**b**) TOCNF and (**c**) TOCNF-AT.

**Figure 6 polymers-16-01060-f006:**
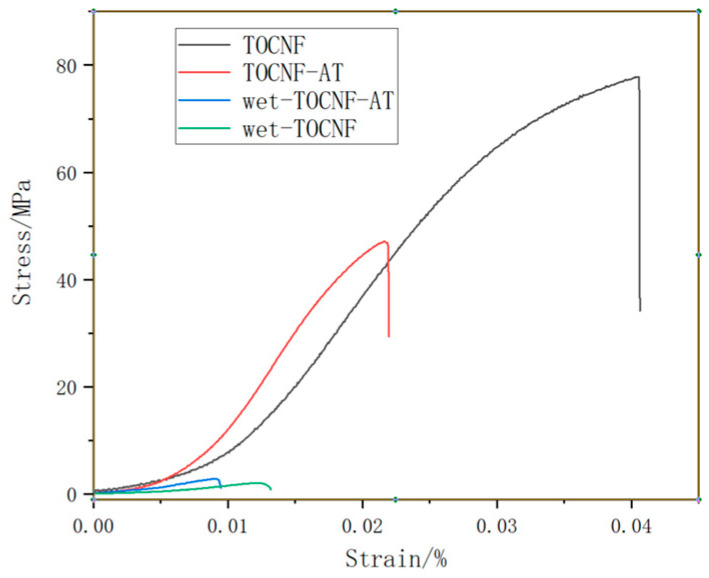
Stress–strain curves of TOCNF and TOCNF-AT under dry and wet conditions.

**Figure 7 polymers-16-01060-f007:**
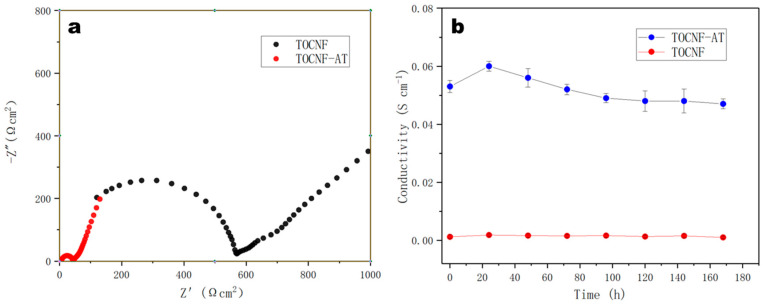
Proton conductivity and stability of TOCNF-AT. (**a**) EIS was used to determine the proton conductivity of the TOCNF and TOCNF-AT. (**b**) Proton conductivity of TOCNF and TOCNF-AT membranes soaked in deionized water for different periods of time.

## Data Availability

The raw data supporting the conclusions of this article will be made available by the authors on request.

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
