# Peer review of "A High-Proton Conductivity All-Biomass Proton Exchange Membrane Enabled by Adenine and Thymine Modified Cellulose Nanofibers"

_polymers, 2024, doi:10.3390/polym16081060_

Round 1

Reviewer 1 Report

Comments and Suggestions for Authors

Xie etal synthesized TOCNF-based proton exchange membranes using an environment friendly procedure. The materials are well  characterized. I have only several minor concerns. 

1. Define TOCNF in abstract?

2. In the experimental section, proton conductivity measurement procedure should be given more clearly such that anyone can reproduce the experiment data. What electrodes were used? How they were aligned with the membranes? 

3. EIS spectra could be compared with that of Nafion as this H+ conducting membrane is reference in this article. Also, elaborated conductivity calculation may be obtained from bode modules of EIS spectra.

Reviewer 2 Report

Comments and Suggestions for Authors

The experimental article “A high proton conductivity all-biomass proton exchange membrane enabled by adenine and thymine modified cellulose nanofibers” is devoted to the strategy for creating proton exchange membranes based on nanocellulose, the surface of which is modified with adenine and thymine. The modification, according to the authors, provided an increase in proton conductivity by 31.2 times and a specific surface area of 6.1 m²/g to 86.5 m²/g. The author's main idea is to create a biodegradable membrane, which in turn can be the basis for fuel cells, supercapacitors and solid-state batteries. A concordance rate of 17% was verified by a reviewer and passed for publication. The list of recommendations is given as a list.

Recommendations:

1. Abstract. It uses the expression “a simple and effective strategy” in relation to the modification of nanocellulose in order to obtain valuable materials, which in turn justifies the scientific novelty of the results. But the authors have already used this expression to prove the scientific novelty of another article, which is now under consideration (Polymers-2937898). There is a recommendation to modify the annotation so that there is no repetition.

2. Lines 46-67. I need to check the text of this paragraph; there is a contradiction in it. On the one hand, the authors claim that “our team previously created proton exchange membranes,” on the other hand, “proton conductivity has not been studied.”

3. Lines 69-86. The text does not contain the statement “creation of a hydrogel,” which is then “specified” in paragraph 2.2 Preparation of CMC-AT nanofiber hydrogel adhesive. We need to fix the error. Authors should carefully proofread the text of the article in connection with the article template (Polymers-2937898).

4. 2.1. Experimental materials. The object of research “nanocellulose” and its characteristics are not indicated here, which the authors plan to oxidize using “TEMPO”.

5. It is necessary to provide evidence of the nanostructure of oxidized cellulose.

6. Section 2.2 should be written in the past tense.

7. Figure 2. Figure caption: you need to exclude the word “cotton” and indicate the object of study, as indicated in the title and text of the article. For Polymers, the terms “nanocellulose fiber” and “cotton” are not identical.

8. Figure 3. Again in the caption “cotton”, and the text discusses the decrease in crystallinity when converting “cotton” to “nanocellulose fiber” (lines 165-166). It is necessary to provide the crystallinity values of three objects: “nanocellulose fiber”, TOCNF and TOCNF-AT.

9. Lines 172-177. Question: Modification of nanofibers with adenine and thymine does not lead to aggregation of nanofibers, as described in the article (Polymers-2937898)? You need to work carefully on this text. There is a contradiction.

10. Lines 186-188 with reference to Figure 4. The meaning of “fiber diameter” given by the authors is absolutely not obvious from the figure. Needs to be fixed. Figures 4c and 4d are also very similar to make strong inferences about their differences.

11. Lines 203-215. Modification of cellulose nanofibrils with adenine and thymine resulted in the formation of large pores. The article (Polymers-2937898) describes that the same modification leads to the formation of a strong adhesive. The author’s “a simple and effective strategy” provides two opposite phenomena. It is necessary to pay more attention to this important fundamental phenomenon, especially since the length of this article allows it (only 10 pages).

12. Section 3.6. Proton conductivity test. A discussion of the proton conductivity of the innovative film compared to the commercial Nafion showed a negative result. It is necessary to continue this theme and develop a strategy to achieve the replacement of synthetic materials with modified cellulose or other products.

13. The conclusions need to be edited and honestly indicate the unattainability of the proton productivity of commercial Nafion.

14. References. The authors actively cite literature in the introduction (27 sources), but then use only 7 references. It turns out that the first 27 sources were needed only to formulate the problem. There is an opportunity to improve this situation.

Reviewer 3 Report

Comments and Suggestions for Authors

In this work, Xie Chong et al. developed a simple and effective strategy to prepare cellulose-based proton exchange membranes that can simultaneously meet sustainability and high-performance standards. Once modified, the proton conductivity of the TOCNF membrane increased by 31.2 times compared to the original mem-brane, with a specific surface area that had risen from 6.1 m²/g to 86.5 m²/g. This work will be helpful to researchers in related fields. Therefore, after addressing the following issues, this manuscript seems worthy of publication in the journal Polymers.

1. The acronym TOCNF (in line 21), EDCI-NHS (in line 71), TOCNF-AT. AFM and SEM (in line 75) are not explained when they show up first time in the paper.

2. Introduction. The authors should clarify the transfer and transformation mechanism of adenine groups in proton conduction, and the advantages of integrating adenine (A) and thymine (T) into nanocellulose fibers in detail in the introduction for readers to understand. Also, provide detailed and informative information about published articles. At the end provide the importance of the study and objectives selected for the study.

3. Figure 1. Schematic diagram of cellulose nanofiber modification and proton conduction. The author should divide it into (a) cellulose fiber (b) TOCNF (c) TOCNF-AT in the title and explain it in the text. Because Figure 1 cannot be found in the text.

4. Page 3, line 94. 2.2. Preparation of CMC-AT nanofiber hydrogel adhesive (The entire paragraph has no content, is it missing?)

5. In addition, Page 3, line 95. 2.2. Preparation of TOCNF and TOCNF-AT proton exchange membranes is 2.3.? It is difficult to understand, please reshape it.

6. Figure. 2、Figure. 2、Figure. 5、Figure. 6、Figure. 7 has weak quality.  Please improve. (The X and Y axis fonts of the graph need to be adjusted.)

7. 3.1. Fourier transform infrared absorption spectroscopy analysis should correspond to Figure 2, but the entire text does not indicate that it is derived from the results of Figure 2.

8. Page 4, line 122-124. The crystal structures of CMC and CMC-AT were observed by X-ray diffraction (XRD) spectroscopy. It should be measured Figure 3. X-ray diffraction (XRD) spectra of cotton, TOCNF and TOCNF -AT. Please explain and correct.

9. Page 4, line 125-128. The mechanical properties of the prepared CMC and CMC-AT films were evaluated using a universal testing machine (UTM) Tensilon RTC 250A, A&D Company Ltd. Surface area and pore size distribution are tested by BET (ASAP , Micromeritics). It should correspond to Figure 5. TOCNF and TOCNF-AT films. Please explain and correct it.

10. Conclusion - The authors conclude by writing that "This study paved a new path for the preparation of environmentally friendly mem-brane materials that could replace the commonly used non-degradable ones, highlighting the potential of nanocellulose fiber membrane materials in sustainable applications such as fuel cells, supercapacitors, and solid-state batteries". It is impossible to express the main point of this article. Conclusion should be enriched with an outlook devoted to challenges facing TOCNF membrane and future development directions. Please reshape it.

Round 2

Reviewer 2 Report

Comments and Suggestions for Authors

The experimental article “A high proton conductivity all-biomass proton exchange membrane enabled by adenine and thymine modified cellulose nanofibers” has undergone changes following the recommendations of reviewers. Almost all my recommendations have been implemented. But please pay attention to the captions to Figures 2 and 3: the word “cotton” needs to be replaced, as the authors promised in their Replies to the reviewer.

Author Response

Thank you for pointing out the error. The "cotton" in the legend of Figure 2 and figure 3 has been replaced with "cellulose fibers"

Reviewer 3 Report

Comments and Suggestions for Authors The authors made substantial improvements to this article. The manuscript can be accepted for publication in its current form.    

Author Response

Dear Reviewer,

Thank you very much for your thorough review and positive feedback on our manuscript. We are honored to learn that you consider the improvements substantial and support the acceptance of our paper for publication.

We sincerely appreciate your time and expertise. We look forward to the publication of our research in your journal and hope to continue contributing to the field.